# Clinical Management of Moyamoya Patients

**DOI:** 10.3390/jcm10163628

**Published:** 2021-08-17

**Authors:** Isabella Canavero, Ignazio Gaspare Vetrano, Marialuisa Zedde, Rosario Pascarella, Laura Gatti, Francesco Acerbi, Sara Nava, Paolo Ferroli, Eugenio Agostino Parati, Anna Bersano

**Affiliations:** 1Cerebrovascular Unit, Fondazione IRCCS Istituto Neurologico Carlo Besta, 20133 Milan, Italy; isabella.canavero@istituto-besta.it; 2Department of Neurosurgery, Fondazione IRCCS Istituto Neurologico Carlo Besta, 20133 Milan, Italy; ignazio.vetrano@istituto-besta.it (I.G.V.); francesco.acerbi@istituto-besta.it (F.A.); paolo.ferroli@istituto-besta.it (P.F.); 3Neurology Unit-Stroke Unit, Azienda Unità Sanitaria Locale-IRCCS di Reggio Emilia, 42122 Reggio Emilia, Italy; Marialuisa.Zedde@ausl.re.it; 4Neuroradiology Unit, Azienda Unità Sanitaria Locale-IRCCS di Reggio Emilia, 42122 Reggio Emilia, Italy; rosario.pascarella@ausl.re.it; 5Cellular Neurobiology Laboratory, Fondazione IRCCS Istituto Neurologico Carlo Besta, 20133 Milan, Italy; laura.gatti@istituto-besta.it; 6Experimental Microsurgical Laboratory, Fondazione IRCCS Istituto Neurologico Carlo Besta, 20133 Milan, Italy; 7Cell Therapy Production Unit, Fondazione IRCCS Istituto Neurologico Carlo Besta, 20133 Milan, Italy; sara.nava@istituto-besta.it; 8Neurorehabilitation Department, IRCCS Istituti Clinici Scientifici Maugeri, 20138 Milan, Italy; eugenio.parati@icsmaugeri.it

**Keywords:** moyamoya angiopathy, moyamoya disease, moyamoya syndrome, diagnosis, management, therapy, revascularization

## Abstract

Moyamoya angiopathy (MMA) is a peculiar cerebrovascular condition characterized by progressive steno-occlusion of the terminal part of the internal carotid arteries (ICAs) and their proximal branches, associated with the development of a network of fragile collateral vessels at the base of the brain. The diagnosis is essentially made by radiological angiographic techniques. MMA is often idiopathic (moyamoya disease-MMD); conversely, it can be associated with acquired or hereditary conditions (moyamoya Syndrome-MMS); however, the pathophysiology underlying either MMD or MMS has not been fully elucidated to date, and this poor knowledge reflects uncertainties and heterogeneity in patient management. MMD and MMS also have similar clinical expressions, including, above all, ischemic and hemorrhagic strokes, then headaches, seizures, cognitive impairment, and movement disorders. The available treatment strategies are currently shared between idiopathic MMD and MMS, including pharmacological and surgical stroke prevention treatments and symptomatic drugs. No pharmacological treatment able to reverse the progressive disappearance of the ICAs has been found to date in both idiopathic and syndromic cases. Antithrombotic agents are usually prescribed in ischemic MMA, although the coexisting hemorrhagic risk should be considered. Surgical revascularization techniques, which are currently the best available treatment in symptomatic MMA, are associated with good long-term outcomes and reduced ischemic and hemorrhagic risks. Given the lack of dedicated randomized clinical trials, current treatment is mainly based on observational studies and physicians’ and surgeons’ expertise.

## 1. Introduction

Moyamoya angiopathy (or arteriopathy; MMA) is a peculiar affection of the cerebral arteries characterized by progressive steno-occlusive lesions of the terminal part and proximal branches of the internal carotid arteries (ICAs), associated with the development of an unstable network of collateral vessels at the base of the brain (moyamoya vessels) [1,2]. These vascular hallmarks are responsible for the main disease clinical features, which are recurrent ischemic and hemorrhagic strokes. The diagnosis is essentially made by radiological angiographic techniques: all disorders with such morphological appearance on imaging could be described as MMA. MMA has been conventionally classified as moyamoya disease (MMD) if the condition is isolated and thus “idiopathic”, whereas moyamoya syndrome (MMS) refers to cases in whom this radiological appearance occurs in association with specific other acquired (i.e., autoimmune diseases) or hereditary disorders (i.e., neurofibromatosis type 1, sickle cell anemia, Down syndrome, Noonan syndrome, Costello syndrome, Alagille syndrome) [1,2].

MMD has been found in all ethnicities, with a marked East-West gradient according to the available studies: the highest incidence is estimated at around 1 per 100,000/year in Japan and about ten times lower in Western countries [3]. However, its incidence is probably underestimated because MMA features may be missed by incomplete instrumental work-up, for example, in acute stroke cases in the emergency setting and in asymptomatic or paucisymptomatic cases. Despite its rarity, the disease is highly impacting, leading the affected children and adult subjects to develop acute and chronic neurological (sensorimotor, speech, cognitive) deficits and progressive physical disabilities. The mortality rate is estimated at around 6.8–28.6% in hemorrhagic cases [4]. Autonomy loss at 5-years follow-up has been observed in 60% of children and 20% of adults [3].

Especially for idiopathic but also for syndromic cases, the pathogenesis of MMA has not been completely elucidated, although angiogenic abnormalities, as well as genetic susceptibility, have been hypothesized to play a role in disease development [5]. The lack of information describing disease pathophysiology and natural history, mainly for the idiopathic forms, and especially in Western countries, has notably limited the development of treatment reversing the occlusive arterial process.

It should be pointed out that the available treatment strategies are essentially shared between idiopathic MMD and MMS, including pharmacological and surgical stroke prevention treatments and symptomatic drugs. Given the absence of randomized trials and guidelines, therapeutic decisions are currently based on clinicians’ and surgeons’ experience and center tools [6]. In addition, as for other rare diseases, insufficient awareness of healthcare providers about this clinical entity bears a notable risk of delayed or wrong diagnoses and missed or inappropriate treatments [7].

To increase and enhance disease knowledge and recognition, especially among physicians from other specialties and/or with different expertise, we elaborated this narrative review to provide a general yet comprehensive dissertation outlining the distinctive features of the disease and the main principles of its diagnosis and treatment. To this effect, the authors’ personal experience was integrated by a literature search performed on PubMed through the keywords “moyamoya” AND “angiopathy”/“arteriopathy”/“disease”/“syndrome”/“epidemiology”/“pathophysiology”/“symptoms”/“diagnosis”/“treatment”/“surgery”/“prognosis”. A large part of the most recent themed literature refers to limited, mostly mono-centric patient populations, however only single case reports, if not emblematic, were excluded from our consideration. Representative references for each aspect of MMA diagnostic and therapeutic work-up have been selected.

Other than the intrinsic limitations of a non-systematic review approach, our aim was to provide a practical overview, a global perspective on the current state-of-the-art for the available strategies for management and treatment of MMA patients, with particular consideration for the Western countries setting, to empower disease recognition and to set a grounding for the appropriate clinical work-up for suspected and confirmed cases, still bearing in mind that the best therapeutic strategy should be tailored for each case.

## 2. Pathophysiology

Several mechanisms have been hypothesized, including inflammation, upregulation of angiogenic factors, and abnormalities of endothelial progenitor cells (EPCs). For a long time, anomalies in angiogenesis and vasculogenesis have been suggested as potential disease mechanisms since altered levels of cytokines, chemokines, and growth factors were detected in cerebrospinal fluid (CSF) and serum of MMA patients [5,8,9,10,11,12,13,14,15]. Afterward, the association of MMA with several heritable conditions (i.e., Down syndrome, sickle cell disease, neurofibromatosis type 1), the high familial rate, and the strong linkage between the disease and variants of Ring Finger Protein 213 (*RNF213*) gene in East Asian patients strengthened the role of genetic factors in MMA pathogenesis [16,17,18]. Thus, the trend of research has drastically changed, and studies focusing on the biological effect of mutant *RNF213* have been developed. However, probably all these pathways contribute to the disease pathophysiology, and MMA results from a complex mechanism in which acquired infectious, inflammatory, and flow dynamic conditions [8,9,19,20] may trigger the disease in genetically susceptible individuals through angiogenic and vasculogenic pathways abnormalities [5].

Moreover, it has been hypothesized that the formation of microaneurysms might be an additional pathophysiologic explanation [21,22]; small aneurysms arising from collateral vessels have been associated with intraventricular hemorrhage [23]. Understanding MMA is challenging and limited by the lack of specific in vivo and in vitro disease experimental models. Nevertheless, several preclinical in vivo or in vitro MMA models, including EPCs, smooth muscle cells (SMCs), iPSC, as well as *RNF213*-KO animal models, or more recently, surgical models, were established to improve our knowledge about disease drivers [12,13,24,25,26]. Regarding the pathophysiology of ischemic cerebrovascular events, it has been demonstrated that endovascular thrombosis, together with hyperplasia of smooth muscle cells, leads to progressive stenosis or occlusion of the distal ICA [2]. Histopathological findings in the carotid terminations include fibrocellular thickening of the intima, irregular waving of the internal elastic lamina, and attenuation of the media, all contributing to the progressive luminal stenosis (“shrinkage”) in the carotid fork. However, it has also been demonstrated that the outer diameter simultaneously decreases, as opposed to atherosclerotic degeneration [27,28].

Moreover, increasing evidence supports the hypothesis that artery-to-artery embolism may also contribute to ischemic events based on microemboli signal (MES) monitoring by transcranial ultrasound, mainly in the early stages of the disease and after recent ischemic stroke [29].

## 3. Clinical Features

The cerebrovascular expression of the disease can be ischemic or hemorrhagic. Ischemic episodes (stroke or transient ischemic attack, TIA) are usually multiple and recurrent due to the steno-occlusive lesions causing acute transient or permanent symptoms in the carotid branches or watershed territories. Ischemic events can be triggered by situations that may increase the hemodynamic demand as physical activity, dehydration, fever, and crying in babies. Hemorrhagic events (intracerebral, intraventricular, or more rarely subarachnoid hemorrhage) are mainly due to the rupture of fragile deep collateral vessels or saccular aneurysms located in the circle of Willis. MMA-related cerebrovascular events vary according to age and ethnicity. In Eastern countries, primarily Japan and Korea, MMA has a bimodal pattern of onset age (5 to 10 and around 40 years), a female prevalence, and hemorrhagic presentation accounts for around 30% of cases [30]. On the contrary, MMA Caucasian adults usually present with TIA or stroke and much more rarely with cerebral hemorrhage [31,32,33]. Children present mainly with ischemic events [32,34,35].

Intellectual disability and cognitive disorders have been described in both children and adults, also in patients who are asymptomatic for cerebrovascular events due to chronic cerebral hypoperfusion [36,37,38,39]. In addition, “indirect symptoms” likely caused by the aberrant collateralization have been described. Migraine-like (with or without aura) and tension-type-like headache attacks have been reported in up to 50–70% in different series of Eastern and Western MMA patients [3,4,27,28,29,30,31,37]. Headache, which has been supposed to be due to collateralization processes and stimulation of dural nociceptors by dilated transdural collaterals, was also associated with brain hypoperfusion [40]. Seizures expressing epilepsy secondary to cortical ischemia are frequent, primarily in pediatric patients [41]. Occasionally, movement disorders may occur in pediatric or adult patients with MMA-related basal ganglia lesions. In particular, chorea, preeminently as onset manifestation in children [42], was also observed in adults and even in the absence of parenchymal lesions, probably due to a hemodynamic impairment affecting the MCA perforator territories [43].

## 4. Neuroradiological Diagnosis

MMA hallmarks are represented by: (1) the progressive stenosis or occlusion of the terminal branches of the ICAs, mainly affecting the proximal portion of the middle and anterior cerebral arteries (MCAs and ACAs) and, less frequently, the posterior cerebral arteries (PCAs), (2) the development of an abnormally fine vascular network (“moyamoya vessels”) in the basal ganglia, which is thought to be produced by the dilation of perforating arteries and by an abnormal formation of collaterals, as compensatory mechanisms in response to the chronic hypoperfusion deriving from the steno-occlusive process (Figure 1).

These features, stated initially by the *Research Committee on Spontaneous Occlusion of the Circle of Willis (moyamoya disease)* in Japan, are accepted worldwide as diagnostic criteria [1,2,44,45,46]. The digital subtraction angiography (DSA) is considered, so far, the diagnostic gold standard, allowing the confirmation of MMD when the abovementioned neuroradiological features are found bilaterally, and other causes of steno-occlusive processes can be ruled out. The main MMA mimickers include large vessel vasculitis, arterial dissections, severe atherosclerosis, and traumatic and radiation sequelae; these conditions are generally defined as “moyamoya-like”, and although the specific treatment of the primary disease is commonly applied, to date, no pharmacological therapy has been found able to producing reversal; in the end, they could benefit from the same management for MMD and MMS. DSA, other than diagnosis, is currently used to assess disease severity, collateral vessel status, and show dynamic changes during the disease course in operated and non-operated cases (Figure 2).

MMD progression has been traditionally categorized into six different stages [47], representing the temporal trend of compensatory mechanisms rather than the severity of the disease (Table 1).

Initially, steno-occlusive changes in the terminal ICAs are observed; the development of moyamoya vessels (depicted by angiographic studies as a “puff of smoke”) occurs at the early stage of MMD (stages I-III), while compensatory development of trans-dural/trans-cranial anastomosis from the external carotid artery (ECA) system and gradual disappearance of moyamoya are the main features of the late-stage (stages IV-V), finally leading to the disappearance of intracranial ICA (stage VI) [48]. This last process, which is known as “ICA-ECA conversion”, represents the ultimate destination of the MMD course, either spontaneous or surgically driven [49], which ideally should be reached without developing ischemic or hemorrhagic symptoms.

Other than DSA, other imaging tools offer useful and less invasive methods to identify the typical neuroimaging features of MMA (Table 2) and evaluate parenchymal damage and cerebral hemodynamics. In particular, the revised diagnostic criteria of MMA [44] stated that a definitive diagnosis could also be made by MR imaging/angiography without DSA:-MRA showing stenosis or occlusion of the terminal portion of the intracranial ICA or proximal portions of the ACA and/or the MCA.-Presence of the abnormal vascular networks near the occlusive or stenotic lesions by MRA or MRI demonstrating two or more flow voids in the basal ganglia on each hemisphere.

In addition, Houkin et al. established MRA grades for evaluating MMA as an alternative to conventional angiography [50], with a scoring system that showed a good correlation with Suzuki’s stages (Table 2).

“Periventricular anastomosis” is a more recent terminology for moyamoya collaterals, which are defined as an anastomosis between the perforating or choroidal artery and medullary artery in the periventricular area. Specific techniques of reformation for CT or MR angiography imaging were developed to facilitate the visualization of such vessels by suppressing the effect produced by other overlapping vessels, and scoring systems have been elaborated. Their presence has been found to be independently associated with the hemorrhagic presentation of MMA [51,52].

High-resolution MR-based imaging techniques have been applied to explore vessel wall features and have demonstrated concentric enhancement on bilateral distal internal carotid arteries and shrinkage of the middle cerebral artery [28,53].

A cerebral MRI can raise diagnostic suspicion and is helpful for MMA patients’ baseline and follow-up assessment. An MRI may be helpful to show acute and old ischemic and hemorrhagic lesions (T1-, T2-, T2*-weighted sequences), microbleeds, and white matter hyperintensities in the distal vascular bed supplied by penetrating branches of MCA/ACA and related to border-zone infarcts. Acute ischemic manifestations can be distinguished based on the distribution pattern in DWI with hemodynamic (7%), embolic (83.7%), and “deep” (9.3%) damage [54]. In addition, an MRI can reveal the so-called “ivy sign”, which is depicted as a result of linear hyperintensities in the sulci and subarachnoid space on fluid-attenuated inversion recovery (FLAIR) or post-contrast T1-weighted images with a continuous or discontinuous pattern [55]. Although it is often considered a typical MMA feature, the ivy sign is not specific and can be found in different steno-occlusive pathologies as an expression of activation and flow engorgement in leptomeningeal vessels [56,57,58,59]. Another peculiar feature of MMA is the “brush sign” due to increased conspicuity of deep medullary veins on susceptibility-weighted imaging (SWI); it possibly represents a predictor of infarction after surgical revascularization and thus of MMA severity [60].

Perfusion studies (CT-perfusion; perfusion-weighted imaging—dynamic susceptibility contrast, PWI-DSC; arterial spin labeling, ASL; single-photon emission computed tomography, SPECT with Tc-99m-HMPAO or challenge with acetazolamide for evaluation of cerebrovascular reactivity, and [(15)O]-water PET) are also commonly used in MMA patients to assess ischemic impairment severity and thus to establish an indication of surgery [2,61,62,63,64,65,66,67,68]. These techniques also enable the evaluation of surgical revascularization efficacy, with perfusion being potentially improved by surgery beyond angiographic findings.

Although each technique has a well-documented field of application, some discrepancies could be found while applying different perfusion techniques in single cases, probably depending on the dynamic course of cerebral hemodynamics and technical issues and specificities. For example, CT perfusion imaging is performed on the premise that all blood flows are distributed from the ICA through the MCA-M1 segment to the whole brain; thus, the development of EC-IC collaterals in MMA patients could determine reduced CBFs studies on CT perfusion but not on SPECT assessment, that could instead reveal an adequate balance of perfusion. Deducing which would be the most informative technique/s for every single case could be tough considering the high heterogeneity and dynamicity of the condition and the absence of apparent clinical correlates. Thus, ideally, a multimodal approach is advisable to assess MMA patients and estimate their risk profile for cerebrovascular complications, to plan timely interventions and follow-up, always integrating the complementary findings provided by different techniques (Table 3). Accordingly, some classification systems integrating clinical, angiographic, and hemodynamic features have been proposed [6].

If the typical angiographic features are found unilaterally, and without identifiable determining causes, the disease is classified as “unilateral” or “probable” MMA [2]. In these cases, progression to bilateral involvement is observed at variable rates, up to 50% of cases [66], and is more frequent and rapid in pediatric patients than in adults. Consequently, patients with unilateral involvement should be regularly followed up with imaging.

Due to the heterogeneous disease presentation, MMA should be suspected in any case of acute or chronic cerebrovascular symptoms, both ischemic and hemorrhagic, especially in young patients and in the absence of vascular risk factors. Asymptomatic patients may be identified incidentally or by imaging performed for screening in the abovementioned clinical conditions that have been reported in association with MMS. Indeed, patients affected by diseases commonly associated with MMA should be screened at least with the less invasive angiographic techniques, even in the absence of specific neurological symptoms. It is also reasonable to recommend screening of first-degree relatives of MMA patients, at least through the less-invasive diagnostic techniques. Other instrumental examinations, such as EEG and neuropsychological tests, are suggested in MMA patients’ assessment to further characterize their clinical phenotype and monitor natural disease history in operated and non-operated patients. Since seizures can be misdiagnosed as cerebrovascular events and vice versa, EEG could have a significant role in differentiating these diagnoses and help clinicians to set the appropriate treatment.

## 5. Medical Treatment

To date, no curative treatment allowing regression of the occlusive arterial lesions has been found for MMA. Although in selected patients, revascularization surgery is the recommended treatment to prevent stroke risk [44], several patients have surgical contraindications or weak or debatable indications. Medical therapy, including antithrombotic therapy, is the standard of care for stroke prevention in patients at high risk of cerebrovascular complications [69], and this might also be applied to MMA patients [2,44]. However, MMA represents a peculiar cerebrovascular disorder with coexisting ischemic and hemorrhagic risk; besides, the ischemic potential is probably due to a hemodynamic mechanism, and the rationale for prescribing antithrombotics could be under debate. Unfortunately, to date, there are only limited reports focused on antiplatelet therapy for MMA, and the lack of randomized controlled trials obliges to very insufficient supporting evidence. Moreover, the use of antiplatelet agents is controversial even in MMA patients with ischemic stroke, in particular for Asians [70], because of their poor efficacy in improving blood supply and potentially increased hemorrhagic effect [71].

Moreover, few studies have reported the efficacy of antiplatelet agents among non-surgical patients or before surgery. A recently published Chinese multicenter retrospective cohort study [72] tried to evaluate the effectiveness and safety of antiplatelet treatment in patients treated conservatively or with surgical revascularization [32,73,74,75,76,77]. Despite some limitations (a large portion of pediatric patients, relatively aged at onset patients, retrospective design), the study concluded that antiplatelet therapy was beneficial for reducing other cerebral ischemic events (5.7 vs. 15.1%). Aspirin was demonstrated to also be effective in the postoperative management of ischemic MMA [78,79]. Other recent Japanese studies demonstrated that antiplatelet agents different from aspirin (clopidogrel and cilostazol) effectively improved cerebral perfusion in adult patients with symptomatically ischemic MMA [80,81]. Moreover, some preliminary results suggest that patients treated with cilostazol show a greater improvement in cognitive evaluation than in those treated with clopidogrel [82].

A recent population study based on the Korean National Health Insurance Service database found that any antiplatelet use was associated with a reduced risk of death (hazard ratio, 0.77; 95% CI, 0.70–0.84) in a multivariate model. Among antiplatelet agents, cilostazol was associated with a more significant reduction of mortality [83]. Moreover, observational studies in Western populations concluded for a benign and stable course of MMA with a ~3.5% annual stroke risk without differences in the clinical outcomes between the operated and the conservative groups [84]. According to this clinically benign course, the same cohort showed a slow progression of MMA-related changes on MRI/MRA [85]. Finally, no evidence is available on the benefit-risk balance of other antithrombotic strategies available in the acute phase, such as intravenous (i.v.) thrombolysis with rtPA, which to date has been described only in case reports [86,87,88]. The limited recommendation in using thrombolysis in MMA is mainly supported by the reported increased risk of hemorrhage and mortality [89]. Considering the unique features of MMA, the balance between ischemic and hemorrhagic risk should be constantly reconsidered over time to identify the best therapeutic strategy that can vary in different settings and timings, with a special concern about using antiplatelets in the perioperative period and after acute ischemic stroke.

### 5.1. Headache Management

Headache is one of the major complaints of MMA patients, and although its actual pathophysiological mechanisms have not been fully characterized, it could be related to cerebrovascular impairment [90]. Pain features can resemble both migraine and tension-type headaches or a combination of the two as well. Most importantly, specific treatment strategies for headaches in MMA patients have not been established. Okada et al. observed that STA MCA anastomosis effectively relieves headaches in patients with ischemic MMA manifesting severe headaches, probably by improving perfusion pressure and cerebral circulation [90]. However, the experience of Seol et al. in the pediatric population suggests that headaches can persist or develop after indirect bypass surgery despite successful prevention of cerebral ischemia [91]. In addition, the progressive collateralization and redistribution of blood flow could be considered a cause of headaches in MMA patients. Headache by itself does not represent an indication for surgery.

Since systematic studies have never addressed symptomatic treatment, drugs are currently administrated according to personal experience and considerations on drug side effects. Caution in the use of NSAIDs derives from their antiplatelet effect (potentially increasing the bleeding risk) and their vasoactive properties (inducing vasoconstriction). On the contrary, some experiences in ischemic onset-type pediatric moyamoya patients who experience headaches and aspirin administration demonstrated effectiveness in alleviating headaches by inhibiting platelet activation [92]. Vasoconstriction could also be induced by triptans (this phenomenon has also been shown in the STA). Among common prophylactic strategies, the impact on systemic blood pressure by β-blockers or calcium channel blockers should also be considered to avoid hypoperfusion [40]. Co-occurrence of headache and epilepsy in MMA patients could drive neurologists to select prophylactic medication with dual efficacy, such as topiramate, lamotrigine, or valproic acid. However, despite these practical considerations, specific studies are lacking.

### 5.2. Epilepsy Management

Epilepsy in MMA could occur:before stroke: the typical epileptogenic focus of MMA is located in the territory of the ICA, probably being an expression of ischemic damage;after stroke: epilepsy in MMA is a recognized type of poststroke epilepsy;post-surgery: the surgical procedure itself, by causing a breach and reorganizing vascular dynamics, might be epileptogenic.

Regarding the latter point, in adult MMA patients, the incidence of delayed postoperative seizure after direct revascularization seems to be higher than that of other pathologies [93]. No evidence about the best antiepileptic drug (AEDs) for the treatment of epilepsy in MMA patients is currently available. MMA epilepsy is preeminently focal and symptomatic. For newly diagnosed focal seizures, currently, carbamazepine and lacosamide are often first-line treatment options; levetiracetam, valproic acid, and lamotrigine can be considered too [94]. Medication tolerability, interactions, and potential pregnancy planning are significant factors to be considered in drug selection. In poststroke epilepsy, despite similar efficacy, lamotrigine was observed to be superior to carbamazepine regarding tolerability; due to a lower rate of drug interactions, levetiracetam is, however, the most commonly prescribed AED [94,95]. Lamotrigine and levetiracetam have a lower fetal risk; particularly, lamotrigine is usually the first choice for fertile women [41].

## 6. Surgical Treatment

Surgical revascularization appears the most effective strategy to improve cerebral hemodynamics, although this specific aspect has not been evaluated in randomized clinical trials. The revascularization goal is to reduce the risk of ischemic events by improving cerebral blood flow (CBF) and restoring cerebral vascular reserve (CVR) through the provision of collateral pathways, based upon the evidence that MMA affects the ICAs and their main branches but spares the ECAs [2]. The Japan Adult Moyamoya (JAM) Trial, the first prospective randomized controlled trial focused on MMA surgical treatment, demonstrated a preventive effect of surgical revascularization against rebleeding, probably due to the post-surgical reduction of moyamoya vessels and hemodynamic stress in fragile collaterals [96]. The main indications for surgical treatment [1] are:recurrent symptoms related to cerebral ischemic mechanisms;cerebral hemodynamic impairment with decreased regional CBF, vascular response, and perfusion reserve seen in hemodynamic neuro-radiological studies;hemorrhage due to the rupture of the posterior collateral vessels (e.g., thalamus or trigone of the lateral ventricle), the so-called “posterior cerebral hemorrhage” [96,97].

Surgical indication for hemorrhagic MMA has been controversial for a long time; however, the JAM trial [96] recently showed a preventive effect of revascularization surgery against rebleeding in patients who experienced intracranial hemorrhage within the preceding year, supporting that surgery should also be performed in hemorrhagic MMA patients. Miyamoto et al. delved into the JAM trial results [96], underlining the effect of bypass surgery in preventing bleeding risk according to the bleeding site, with a more significant benefit from surgery mostly in patients with posterior hemorrhages, with respect to the hemorrhage attributed to rupture of caudate nucleus or putamen collateral vessels (“anterior hemorrhage”) [97]. Special considerations must also be made with pediatric MMA cases since the disease seems more aggressive in children. For this reason, revascularization surgery is indicated for most children, and it should be performed promptly [2,98].

Preoperative DSA is recommended to assess the extension of the occlusion and the collateral network, identify donor arteries, and avoid the disruption of pre-existing collateral vessels possibly developed from ECA. CT or MRI could be used to evaluate possible ischemic and/or hemorrhagic brain lesions. CT angiography can depict the course of STA and peripheral branches of MCA to better plan the surgical procedure.

Surgical techniques can be classified into three groups: direct, indirect, and combined revascularization procedures. Direct revascularization, which has been performed in MMA patients since the 1970s, consists of a direct extracranial-intracranial bypass, which could be an STA to MCA bypass, as performed in most cases, or a middle meningeal artery (MMA) to MCA bypass [99,100,101,102]. Anterior cerebral arteries could also be considered as receiving vessels if severe ischemia is found in the ACA area. The occipital artery (OA) could be used as a donor’s vessel for anastomosis to PCA cortical branches to reinforce posterior cerebral artery blood flow [103]. The donor artery is harvested from the scalp and covered by papaverine to prevent vasospasm. Based on the region of hypoperfusion and the surgeon’s experience, various craniotomies can be performed. After the bypass is created, the blood flow through the bypass pedicle can be measured with intraoperative ultrasonic blood flow probes, visualized by indocyanine green angiography, and quantified by FLOW 800 color maps [104,105].

On the other side, indirect revascularization aims to induce spontaneous development of a new vascular network involving pediculate donor tissues, vascularized by the ECA, placed in direct contact with the brain. Different techniques have been proposed to obtain indirect revascularization, all based on a synangiosis procedure, including encephalodurosynangiosis, encephaloduroarteriosynangiosis, encephalomyosynangiosis, encephaloarteriosynangiosis, encephaloduroarteriomyosynangiosis, encephalodurogaleosynangiosis, omentum transplantation, craniotomy with inversion of the dura, and multiple burr holes without vessel synangiosis [106,107,108,109,110,111,112,113]. The direct bypass could improve cerebral hemodynamics immediately, resolving ischemic events rapidly after surgery, although it could be challenging in pediatric patients since their cortical vessels have a smaller diameter and more fragile walls. Indirect revascularization is technically straightforward; however, the development of the collateral network and the increase in cerebral blood flow usually require 3–4 months. Combined direct and indirect revascularization is performed by most MMA hub centers, taking advantage of contemporary immediately increasing the cerebral blood flow with the direct bypass and inducing improved flow over time with the indirect bypass [Figure 3].

Several studies have shown a good safety profile for surgery and reduced subsequent cerebrovascular events in both adult and pediatric patients after surgical revascularization. Scott et al. documented an excellent long-term prognosis in children treated with indirect revascularization by pial synangiosis, showing how this treatment can halt the clinical deterioration of the disease [114]. Kim et al. reported a favorable clinical outcome in 81% of pediatric MMA cases undergoing surgery in a retrospective analysis of 410 patients, with early diagnosis and timely surgical treatment correlated with better prognosis [115]. A systematic review by Fung et al. concluded that 87% of surgically treated patients had benefited from the revascularization, without significant difference between direct, indirect, and combined techniques [113].

Perioperative complications of surgical treatment for MMA include stroke, infection, and intracranial bleeding. In addition to these, after surgery, ischemic events could be induced by cerebral vasoconstriction due to crying and hyperventilation, especially in pediatric patients. A particular complication of bypass procedures is the cerebral hyperperfusion syndrome, which could lead to transient or permanent neurological deficits, observed in 15–27.5% of patients, primarily adults, undergoing direct bypass surgery [75,102,114,116,117]. Postoperative hyperperfusion, whereas not clinically evident, can be defined as a focal and intense increase of CBF followed by its normalization in subsequent SPECT studies [118]. The onset of neurological deficits, in clinical manifesting hyperperfusion, can vary from immediate to up to ten days after surgery [118,119,120]. Impaired cerebrovascular autoregulation and increased vascular permeability have been implicated in the pathogenesis of such conditions [121]. Therefore, after STA-MCA anastomosis, accurate diagnosis of cerebral hyperperfusion and blood pressure control and considering the severity of hemodynamic compromise in the contralateral and/or remote areas are essential for postoperative management of moyamoya patients [122].

## 7. Prognosis

The natural history of MMD and MMS is not well known, mainly in Western countries, since specific studies are lacking or have not been systematically addressed. Several neuroimaging elements, such as brain infarct at cerebral MRI, posterior circulation involvement, collateral site, and specific features (i.e., ivy sign), as well as decreased cerebrovascular reserve, have been identified as risk factors for stroke recurrence both in operated and non-operated cases. Prognostic assessment in MMA could focus on rough outcome measures, such as ischemic and hemorrhagic strokes and stroke-related mortality (which has been estimated at around 6.8–28.6% in hemorrhagic cases) [4].

In addition, a central role in prognosis evaluation should be represented by quality-of-life assessment, seeing as MMA is often featured by the development of functional and cognitive impairment. The social burden of MMA is essential since the disease leads to progressive functional and cognitive disabilities in children and young adults with loss of autonomy at 5-years of 60% in children and 20% in adults, and a 30% long-term impairment of quality of life, including social adaptation difficulties (i.e., school attendance or regular employment) [3,123,124,125]. Although it is not supported by randomized clinical trials or by clear selection indications, surgical direct and indirect revascularization is so far the only available treatment used to reduce incident ischemic and hemorrhagic strokes and improve cognitive deficits in pediatric patients, whereas in the adult population, the data are less clear [126,127,128]. Ha et al. examined long-term postoperative outcome in pediatric MMA patients undergoing indirect bypass, providing satisfactory improvement [129]: the overall clinical outcome was favorable in 95% of the patients (mean follow up was 12 years), with minimal annual risks of symptomatic infarction and hemorrhage and maximal (>99%) 10-year event-free survival rates. Abhinav et al. confirmed the positive role of revascularization, which can decrease the re-hemorrhage rates in hemorrhagic MMA from the 32 to 61% observed in non-operated cases to 12 to 17% [130]. Finally, the outcome could be improved by identifying and proper management of potentially detrimental but modifiable risk factors for postoperative ischemia, such as hyperhomocysteinemia [131] and diabetes [132].

## 8. Summary

MMA represents a heterogeneous and largely unknown disorder. The extreme clinical variability, the heterogeneity of findings on diagnostic assessment, and the small sample size of reported series (mostly in Western countries) limit our knowledge on disease course and the identification of disease progression drivers. MMA diagnostic work-up is not established and shared between centers. DSA is still mandatory for disease diagnosis and is applied to characterize the main MMA features and collateral distribution. Cerebral MRI, which may represent the first step of diagnostic suspicion, is increasingly used to detect symptomatic or silent cerebrovascular events and for patients’ follow-up. The other diagnostic tools, including perfusion studies or advanced MRI techniques, are discretionally applied, based on clinicians’ experience, but no guidelines are currently applied.

In addition, the scarcity of studies on MMA pathogenesis and the lack of animal models have limited the development of specific treatments targeting the progressive stenotic and aberrant collateralization mechanisms.

To date, the best therapeutic option able to reduce the risk of cerebrovascular events is revascularization surgery. However, although meta-analyses would support the role of surgical revascularization in preventing cerebrovascular event risk, indications for surgery are still unclearly defined and influenced by the retrospective design of most included studies. The heterogeneity in protocols for patients’ selection, diagnostic work-up, and surgical techniques, due to the variable experience and resources of centers, represent a major obstacle for the significance of the available studies.

In addition, MMA medical treatment cannot rely on evidence-based guidelines and, so far, is based on anecdotal reports or limited case series. Thrombolysis in the acute phase of ischemic MMA would seem to be contraindicated due to the high hemorrhagic risk. Long-term oral antiplatelet therapy is commonly administered in children and adults with or without a history of ischemic stroke to prevent thrombosis and thromboembolism linked to arterial stenoses. Symptomatic treatments are used to manage migraines, seizures, or movement disorders, although specific recommendations are lacking. Moreover, the disease heterogeneity, the comorbid conditions, and the lack of reliable biomarkers prevent a reliable evaluation of treatment efficacy.

Lastly, despite the disease being severe and may induce motor, cognitive sequelae in several patients, data on MMA natural history are lacking, mostly in Western countries. Although some factors, such as age, comorbidities, and cerebral hemodynamic impairment, have been recognized as potential outcome predictors, no standardized risk stratification systems are available, and therapeutic decisions are based on clinicians’ and surgeons’ experience [6]. For this reason, new scoring systems have been developed and are currently being validated. The Berlin MMD grading system stratifies preoperative hemispheric symptomatology and correlates with postoperative ischemic events [133]. Another scoring system assessing collateral status to predict postoperative outcome and prognosis has been recently proposed [134].

Briefly, many items about the various aspects of MMA management should be further explored by specific research questions. However, the rarity and the extreme complexity of the disease represent the main factors preventing both a timely diagnosis and a “standardized” evaluation of diagnostics and treatments. Our narrative review is primarily meant to highlight the main aspects of MMA with the principle aim to increase awareness about this condition among healthcare providers from other specialties in order to enhance its prompt recognition and proper management, to fight misdiagnoses and inspire the structure of further, more specific studies.

In conclusion, recommendations and evidence-based guidelines for disease management are still largely lacking. Diagnostic and therapeutic pathways are site-based and highly variable, and complete clinical-instrumental work-ups are available only in a few referral centers [135]. For this reason, many MMA patients cannot get access to adequate care processes, finally leading to the increased rate of cerebrovascular episodes, hospitalizations, disability, and mortality, globally increasing healthcare system costs. Therefore, there is an urgent need to provide more clear indications for MMA management. Design and implementation of proper randomized controlled trials to assess the efficacy of different treatment strategies and their impact on the clinical outcome would add essential information.

## Figures and Tables

**Figure 1 jcm-10-03628-f001:**
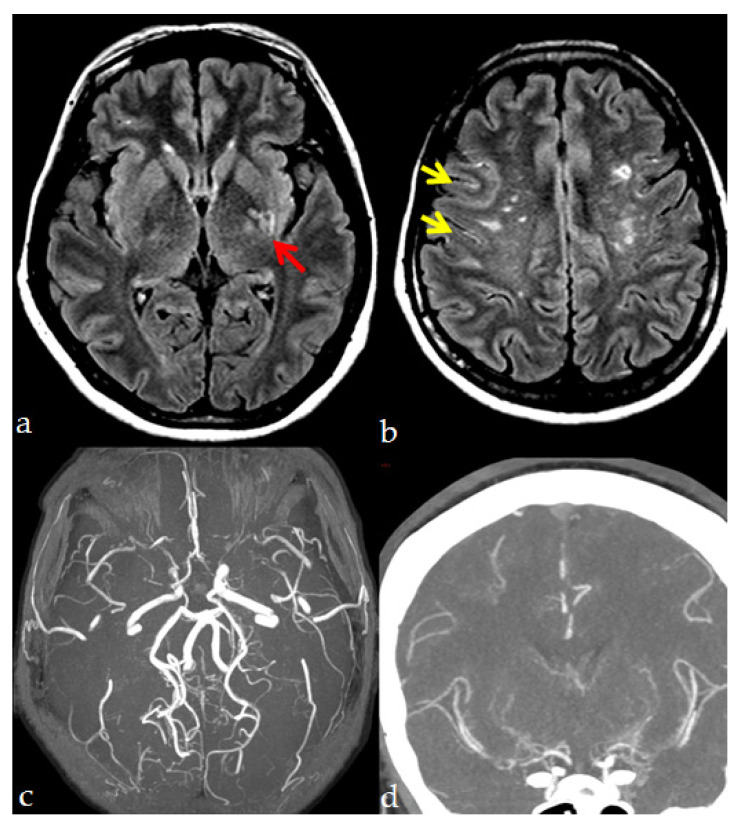
MRI and CT imaging assessment in an MMA patient. Axial FLAIR images show (**a**) a small lacunar infarction in the left posterior lenticular region, which is pointed out by the red arrow; (**b**) multiple, non-confluent, white matter hyperintensities with subcortical distribution in both hemispheres; yellow arrows indicate the Ivy sign. (**c**) MRA with bilateral terminal ICA steno-occlusion with lack of signal in both M1 MCA. (**d**) CTA in coronal MIP showing the same finding as in (**c**) with visualization of the distal sylvian segment of MCA and prominent collateralization in the lenticulostriatal perforator vessels. (Imaging performed at the Neuroradiology Department, AUSL Reggio Emilia; pictures are reproduced with patient’s permission).

**Figure 2 jcm-10-03628-f002:**
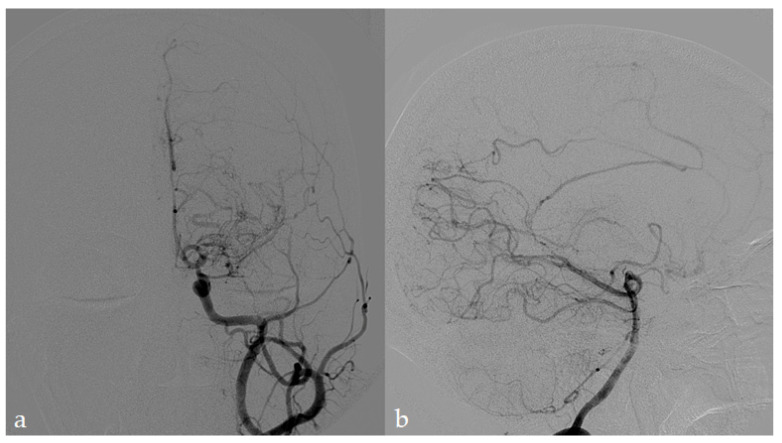
DSA in an MMA patient (same patient of Figure 1). The left ICA (**a**) depicts high-grade stenosis of terminal ICA and proximal M1 MCA segment with moyamoya collateralization pattern. Examining the left VA (**b**), cortical collaterals from PCA and ACA (callosal arcade) to superficial MCA territory are visible. (Imaging performed at the Neuroradiology Department, AUSL Reggio Emilia; pictures are reproduced with patient’s permission).

**Figure 3 jcm-10-03628-f003:**
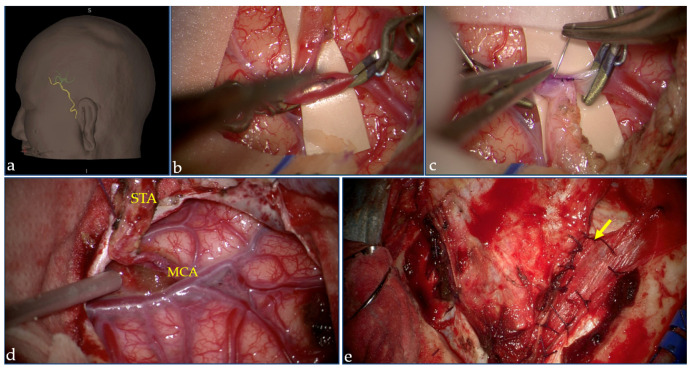
Direct and indirect revascularization surgery. During surgery, a three-dimensional rendering obtained by CT angiography (**a**) shows the course of STA (in yellow) and of the peripheral branches of MCA (green) on the left hemisphere in a patient who underwent combined revascularization surgery. The intraoperative pictures acquired under microscopic view depict the different phases of surgical procedure: in (**b**), temporary clips are positioned in the M4 branch. Then, the STA-MCA bypass is performed with interrupted suture (**c**). After the bypass is completed (**d**), the encephaloduromyosynangiosis with dura and temporal muscle (arrow) is then performed (**e**). (Procedure performed at the Neurosurgery Department, Fondazione IRCCS Istituto Neurologico Carlo Besta, Milan; pictures are reproduced with patient’s permission).

**Table 1 jcm-10-03628-t001:** Suzuki grading system adapted from [47].

Stage I	Narrowing of Terminal ICA
Stage II	Initiation of moyamoya vessels in basal carotid circulation, dilation of intracerebral arteries
Stage III	Intensification of moyamoya vessels, severe carotid stenosis, defection of ACA and MCA
Stage IV	Minimization of moyamoya vessels, defection of PCA
Stage V	Further reduction of moyamoya, disappearance of major cerebral arteries
Stage VI	Disappearance of moyamoya collaterals and ICA, cerebral blood supply comes from external carotid arteries via leptomeningeal anastomoses

ICA: Internal Carotid Artery; ACA: Anterior Cerebral Artery; MCA: Middle Cerebral Artery; PCA: Posterior Cerebral Artery.

**Table 2 jcm-10-03628-t002:** Houkin’s MRA score and grading [50].

Houkin’s Score			Houkin’s Grading	
Main Artery	Findings	Score	Score	Grade
**ICA **	Normal	0	0–1	1
	C1 stenosis	1		
	Discontinuity of C1 signal	2		
	Invisible	3		
**MCA**	Normal	0	2–4	2
	M1 stenosis	1		
	Discontinuity of M1 signal	2		
	Invisible	3		
**ACA**	Normal A2 and its distal signal	0	5–7	3
	A2 and its distal signal decrease or loss	1		
	Invisible	2		
**PCA**	Normal P2 and its distal signal	0	8–10	4
	P2 and its distal signal decrease or loss	1		
	Invisible	2		

**Table 3 jcm-10-03628-t003:** Multimodal imaging assessment of MMA.

*Technique* Focus	Advantages (A)/Disadvantages (D)
***Brain CT*** Brain parenchyma damage	A: Easily accessible in the acute phase; Short acquisition time D: Poor spatial resolution; No information about vessels
***CT Angiography*** Vessel imaging	A: Non-invasive technique; Good spatial resolution; Short acquisition time; Widely available D: Radioexposition; Contrast administration; Poor temporal resolution (without dynamic acquisition)
***CT Perfusion*** Cerebral perfusion	A: Good temporal and spatial resolution; Short acquisition time; Acetazolamide can also be used to assess the CerebroVascular reactivity D: Whole-brain perfusion technology not widely available; Radioexposition; Potential underestimation of CBF in patients with EC-IC collaterals
***Brain MRI*** Brain parenchyma damage	A: Non-invasive; Very good tissue and spatial resolution D: Magnetic field limitations; Claustrophobia; Long acquisition time
***DSC-MRI*** Cerebral perfusion	A: Non-invasive; No exposure to ionizing radiation D: Requires contrast administration; Not fully standardized; Extensive collaterals can prolong arterial transit delays (causing inaccurate assessment of perfusion)
***ASL-MRI*** Cerebral perfusion	A: Non-invasive; No exposure to ionizing radiation or contrast administration; Easy assessment and performance on children D: Not fully standardized; Extensive collaterals can prolong arterial transit delays (causing inaccurate assessment of perfusion); Drug challenge (acetazolamide) with potential side effects
***MR Angiography*** Vessel imaging	A: Non-invasive; No contrast administration; Good spatial resolution in first and second-degree branches D: Relatively long acquisition time; Motion artifacts
***Vessel Wall Imaging*** Vessel wall inflammation or remodeling	A: Differential diagnosis from other steno-occlusive diseases (vasculitides, atherosclerosis, dissections, etc.) D: Not validated for follow-up
***DSA*** Vessel imaging	A: High spatial resolution; High temporal resolution with hemodynamic evaluation; Gold standard for vessel disease D: Invasive; Contrast administration
***Transcranial (Color-Coded) Duplex Ultrasound*** Hemodynamicscerebrovascular reactivity and reserve	A: Non-invasive; Bedside executable; Repeatable; Low cost D: Dependent on acoustic window quality; Diagnostic and grading criteria are not validated in MMA; Operator-dependent
***15O-PET*** Cerebral hemodynamic statuscerebrovascular reactivity and reserve (15O-water PET)	A: Non-invasive; Quantitative measurement of hemodynamic impairment; Useful for follow-up D: Long acquisition time; Not widely accessible; Highly expensive; Radioexposition

Abbreviations (alphabetical order): ASL: Arterial Spin Labeling; CT: Computer Tomography; DSA: Digital Subtraction Angiography; DSC: Dynamic Susceptibility Contrast; MR/MRI: Magnetic Resonance/MR Imaging; 15O-PET: Positron Emission Tomography.

## Data Availability

Not applicable.

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
