# Peer review of "Clinical Management of Moyamoya Patients"

_jcm, 2021, doi:10.3390/jcm10163628_

Round 1
Reviewer 1 Report
The authors revised the manuscript according to my comment.
Two following points should be corrected;
Moyamoya disease, syndrome, arteriopathy is commonly abbreviated to be "MMD", "MMS" and "MMA".
Unnecessary footnotes such as CBF, CBV, and OEF in Table 3 should be deleted.
Reviewer 2 Report
This paper is a review of clinical management of moyamoya disease. Authors cited many articles and seems to be an important article. However, there are some questions.
The major question is that it is not clear how the cited papers were selected. The criteria for citing the articles should be clarified. It is not explained how the articles for this review were selected. As it is a review of the literature, the authors need to explain the selection procedure. The authors should use the PRISMA checklist.
Regarding image findings, there is no mention of periventricular anasmotosis1 or shrinkage of carotid forks2, which have been attracting attention recently. Authors should mention to these issues and should cite appropriate articles.
Intraoperative images are presented in figure, but images of surgery performed at which facility are not specified. It is not specified whether the patient's informed consent has been obtained.
In summary, the purpose of this article is not clear, the criteria for selection of the cited articles are not clear, and it cannot be said to be an appropriate review article. It should take the form of a systematic review.
1 Funaki T, Takahashi JC, Yoshida K, Takagi Y, Fushimi Y, Kikuchi T, Mineharu Y, Okada T, Morimoto T, Miyamoto S. Periventricular anastomosis in moyamoya disease: detecting fragile collateral vessels with MR angiography. J Neurosurg. 2016 Jun;124(6):1766-72. doi: 10.3171/2015.6.JNS15845. Epub 2015 Nov 27. PMID: 26613176.
2 Kuroda S, Kashiwazaki D, Akioka N, Koh M, Hori E, Nishikata M, Umemura K, Horie Y, Noguchi K, Kuwayama N. Specific Shrinkage of Carotid Forks in Moyamoya Disease: A Novel Key Finding for Diagnosis. Neurol Med Chir (Tokyo). 2015;55(10):796-804. doi: 10.2176/nmc.oa.2015-0044. Epub 2015 Sep 15. PMID: 26369872; PMCID: PMC4663029.
Round 2
Reviewer 2 Report
The authors have answered most of my suggestions, and manuscript has been modified accordingly.
It is stated that the patient's informed consent was obtained by using the patient's image, but it is not specified whether the approval of the ethics committee of the research institution has been obtained. The ethical review number should be specified.
Author Response
We thank the Reviewer for pointing out this other inaccuracy. Accordingly, we have now added the section "Ethics" with details at the end of the manuscript:
"The review was performed using standard literature research tools. Representative pictures of moyamoya patients' diagnostic investigations and procedures were selected from the population included in the multicentric, observational GE-NO-MA study. The study design was approved by the Ethics Committee of the Fondazione IRCCS Istituto Neurologico “C. Besta” of Milan (report no. 12, 10/01/2014) and was performed in accordance with the 2013 WMA Declaration of Helsinki. Patients' images have been anonymized and reproduced free from identifying details after obtaining written informed consent from each subject, allowing their use for teaching and scientific purposes". We thank again the reviewer for his careful evaluation of the manuscript and precious suggestions.This manuscript is a resubmission of an earlier submission. The following is a list of the peer review reports and author responses from that submission.
Round 1
Reviewer 1 Report
As a whole, I think it is a very well-written review article. Please let me indicate some points for making it better.
①The most noticed is that the authors use terminology of “moyamoya arteriopathy” through all sentences. I think this review is a review for “moyamoya disease”. I think terminology should be changed to “moyamoya disease” in a unified manner. What do the authors think?
②There is a description that moyamoya disease with hemorrhage onset is rare in section 3, but in Japan, 30-40% of cases of moyamoya disease with adult onset are hemorrhage onset. Review the relevant literature and modify the description.
③It is stated in section 4 that Suzuki’s Stage 5 is ideal, but I think it is not correct understanding. I believe it is right to understand that the idealized EC-IC conversion is to raise Suzuki’s Stage without developing ischemic or hemorrhagic symptoms. Please review again.
④CT perfusion imaging is considered useful as imaging procedures for moyamoya disease in section 4, but CT perfusion imaging is performed on the premise that all blood flows are distributed from the IC through the M1 to the brain, so it is not correct for the condition of patients with moyamoya diseases in which EC-IC collaterals are developed. I have experienced some patients with moyamoya diseases in whom SPECT showed improved blood flow, but CT perfusion images may show reduced CBFs. Consider erasing the CT perfusion image description from section 4.
⑤With regard to section 5, moyamoya disease is a lesion that causes not only ischemia but also hemorrhage, so the use of antiplatelet agents may be used carefully in the perioperative period and after cerebral infarction, with time limits. Some centers may not use antiplatelet therapy for moyamoya disease, which is hemodynamic cerebral ischemia, as it is ineffective. In the current description, readers of the article might misunderstand that the patients with moyamoya disease who develop ischemia should inevitable continue antiplatelet drugs forever. Please review the description again.
⑥Regarding section 5, I think carbamazepine and lamotrigine are not always the first choice. Do you think levetiracetam is most commonly used in clinical practice? Please review the literature again.
⑦Regarding section 6, the description of posterior hemorrhages is difficult to be understood. Consider additional descriptions of what cases of JAM trial could be expected to be effective in preventing hemorrhage.
⑧After direct revascularization for moyamoya disease, hyperperfusion and watershed shift may occur, and various complications due to CBF instability may occur during about 2 weeks from the surgery untill the cerebral blood flow stabilizes. This is very important aspect in the treatment of patients with moyamoya disease, so please consider additional sections.
Reviewer 2 Report
The authors made review article for the management of moyamoya disease. The article is generally well written. I listed the following points to be revised before publication.
MAJOR POINTS
The word “Moyamoya arteriopathy (MA)” is uncommon. No article the authors cited use this word, most of them used “Moyamoya disease (MMD)” and a few articles used “Moyamoya vasculopathy” and “Moyamoya angiopathy”. Also, whether this word means sporadic Moyamoya disease or both sporadic and secondary (or quasi) Moyamoya disease is unclear. It is better to use commonly used word, and clearly state the definition. “Summary” starts with the word “Moyamoya disease” and the word usage is inconsistent in this article. It seems that majority of the articles reviewed refer to only soporadic Moyamoya disease, and better to use the word “Moyamoya disease (MMD)” instead.
Whether the authors target to review the whole world or target the management in the Europe is sometimes unclear and seems to change in each subsection. Clearly state this point in Introduction.
MINOR POINTS
Page 2. The word “usual instrumental work-up” is ambiguous. Did the authors refer to computed tomography? It depends on the country but it is common to obtain MRI/MRA for the evaluation of acute ischemic stroke, and many patients can be diagnosed as MMD by that.
Page 2. The name of the gene (i.e. RNF213) should be presented as Italic.
Page 2. “saccular aneurysms located in the circle of the Willis”. This statement needs citation. Also, the authors should refer to microaneurysms in MMD with citation of some articles (example: https://doi.org/10.3174/ajnr.A4786; DOI: 10.1161/01.str.11.4.405)
Page 3. “adults usually present with TIA or stroke and - more rarely 98 - cerebral hemorrhage”. This might be true in Europe, but not true in Asian countries. When presenting epidemiology of MMD, the authors should clearly define the regional difference.
Page 3. This reference should be included in the same point of Ref. 26 and 27 (10.1161/STROKEAHA.118.022367; 10.1227/NEU.0b013e3182320d1a)
Page 3. “Migraine-like (with or without aura) and tension-type-like headache attacks have been reported in up to 70% of MA patients.” This statement needs citation. Also in which country such a high percentage is reported?
Figure 1. The authors stated “Non-invasive imaging” but CTA cannot be called noninvasive.
Table 1. It is better to present Houkin’s MRA Stage as well as Suzuki DSA Stage in this or different Table.
Explanation of “MA diagnosis” is too long, and the revised criteria (MRA diagnosis) in Page 5 is difficult to follow.
Page 6. “These technique enables also” should be “These technique also enables”
Table 2. “Vantages/Disadvantages” is uncommon usage in English. “Advantages/disadvantages” or “pons and cons” will sounds better.
Brain CT can diagnose flow voids of basal ganglia if they are ovid.
Better not to mix DSC and ASL. DSC requires contrast administration. DSC can measure CBV, CBF, MTT and TTP, but not ASL. ASL can measure CBF, ATT, and if given acetazolamide, CVR. Both contrast agent and acetazolamide carries some risk and should be refer to the disadvantageous points.
PET should be clearly stated as “15O-PET”. “Cerebral vasoreactivity and reserve” is inaccurate because 15O-gas PET measures CBF, CBV, OEF and CMRO2, and 15O-water PET with acetazolamide measures CVR.
No reference is added to the section starts with “Perfusion studies”. Also, 15O-PET is presented in the Table but not in the main text. Each modalities should include some references, such as these (10.1177/0271678x16636393; 10.1136/jnnp.2003.025049; 10.1038/sj.jcbfm.9600187)
Page 9. The following reference showing cilostazol effect to improve cognitive function from the same group of Reference 60 and 61 should be cited (DOI: 10.1080/01616412.2019.1580455).
Page 9. In the headache management section, the following article should be referred (DOI: 10.1007/s00381-020-04991-y).
Page 9. Epilepsy in MMD should cite this article (10.1227/NEU.0b013e31820c045a)
Page 9. The word “posterior cerebral hemorrhage” is not kind to the readers who is not familiar to the JAM trial.
Page 10. “Acute stroke is associated with a higher risk for perioperative complications as further stroke and hyperperfusion syndrome”. This statement needs citation.
Please refer to the cognitive outcome of the surgical revascularization with these articles cited (10.1161/strokeaha.116.016028; 10.1136/jnnp-2019-321069; 10.1007/s12149-020-01473-8)
Round 2
Reviewer 1 Report
I have read the modified article, but I have found that the authors did not agree to the modification in terms of the several points that I considered very important.
Major
Although I have read the authors arguments, I still cannot agree with the use of the term MA. For example, it has been described in Abstrct as "To date, MA etiology and pathophysiology are not fully understand," but this description is incorrect because the cause is obvious for moyamoya syndrome due to arteriosclerosis, radiotherapy, and thyroid dysfunction, which I believe are classified as Mas in the present article. Other expressions are scattered in the sentences where there are concerns that MA may lead to loss of accuracy.
Although I have thoroughly read both additional cited papers on CT perfusion images the authors additionally cited, there is no convincing description of how they process flow other than flow through the M1, which I firstly concerned, and the authors have not made any discussion in this regard. I think there are not some cases that are not suitable for CT perfusion images, but there are some cases that are suitable for CT perfusion images.